# Head and Neck Squamous Cell Carcinoma Biopsies Maintained Ex Vivo on a Perfusion Device Show Gene Changes with Time and Clinically Relevant Doses of Irradiation

**DOI:** 10.3390/cancers15184575

**Published:** 2023-09-15

**Authors:** Victoria Green, Lydia Baldwin, James England, Gayle Marshall, Lucy Frost, Craig Moore, John Greenman

**Affiliations:** 1Centre for Biomedicine, Hull York Medical School, Faculty of Health Sciences, University of Hull, Hull HU6 7RX, UK; lydia.baldwin@hull.ac.uk (L.B.); j.greenman@hull.ac.uk (J.G.); 2Department of Otorhinolaryngology, Head and Neck Surgery, Hull University Teaching Hospitals NHS Trust Hull, Hull HU16 5JQ, UK; james.england3@nhs.net; 3Medicines Discovery Catapult Ltd., Alderley Park, Alderley Edge, Cheshire SK10 4TG, UK; gayle.marshall@md.catapult.org.uk (G.M.); lucy.frost@md.catapult.org.uk (L.F.); 4Medical Physics Service, Hull University Teaching Hospitals NHS Trust Hull, Hull HU16 5JQ, UK; craig.moore3@nhs.net

**Keywords:** HNSCC, perfusion, nanostring, biomarker, irradiation treatment

## Abstract

**Simple Summary:**

The desire to reduce the use of animals in research and drug testing has led to an upsurge in the development of 3-Dimensional models, which try and mimic parts of the human body. However, how these models truly mimic the human situation remains to be fully elucidated. The authors have developed technology (perfusion device) that enables small pieces of human tissue to be maintained outside of the body, enabling the investigation of the effects of various treatments on the patient’s own tissue. The current study describes how this technology has been used to study the gene changes occurring in the tissue, whilst being maintained on the perfusion device, and also the effects of irradiation, providing a deeper understanding of how the patient tissue behaves once it is removed from the body and whether this model will be useful for future treatment testing.

**Abstract:**

Advancements in 3-Dimensional (3D) culture models for studying disease have increased significantly over the last two decades, but fully understanding how these models represent in vivo still requires further investigation. The current study investigated differences in gene expression between a baseline sample and that maintained on a tissue-on-chip perfusion device for up to 96 h, with and without clinically-relevant doses of irradiation, to allow differentiation of model and treatment effects. Tumour tissue samples from 7 Head and Neck Squamous Cell Carcinomas (HNSCC) patients were sub-divided and either fixed immediately upon excision or maintained in a tissue-on-chip device for 48 and 96 h, with or without 2 Gray (Gy) or 10 Gy irradiation. Gene expression was measured using an nCounter^®^ PanCancer Progression Panel. Differentially expressed genes between pre- and post-ex vivo culture, and control and irradiated samples were identified using nSolver software (version 4.0). The secretome from the tumour-on-chip was analysed for the presence of cytokines using a Proteome Profiler™ platform. Significant numbers of genes both increased (*n* = 6 and 64) and decreased (*n* = 18 and 58) in expression in the tissue maintained on-chip for 48 and 96 h, respectively, compared to fresh tissue; however, the irradiation schedule chosen did not induce significant changes in gene expression or cytokine secretion. Although HNSCC tissue maintained ex vivo shows a decrease in a large proportion of altered genes, 25% and 53% (48 and 96 h) do show increased expression, suggesting that the tissue remains functional. Irradiation of tumour tissue-on-chip needs to be conducted for longer time periods for specific gene changes to be observed, but we have shown, for the first time, the feasibility of using this perfusion platform for studying the genomic response of HNSCC tissue biopsies.

## 1. Introduction

Head and neck cancer is ranked the seventh most common cancer worldwide [1], with 12,000 people in the UK diagnosed each year [2] and 4078 deaths in the UK in 2018 (2% of total cancer deaths) [3]. Ninety percent of cases of head and neck cancer arise in the squamous epithelium, Head and Neck Squamous Cell Carcinomas (HNSCC) [4], with tobacco use, alcohol consumption and infection with Human Papilloma Virus (HPV) being the main risk factors, either alone or in combination [5]. HNSCC is more frequently found in men, with the incidence rate ratio being 2.7 (men:women), and those in lower socioeconomic groups at the greatest risk [6]. Treatment commonly involves a combination of surgery and/or chemoradiotherapy, with the most frequently used drugs being docetaxel, cisplatin and 5-fluorouracil [7]. Despite improvements in treatment regimens, such as the introduction of intensity modulated radiotherapy and image guided radiotherapy, which are more targeted ways of delivering radiation doses [8], the survival rate of HNSCC patients has improved very little over recent decades, with 5-year survival rates of 61%, 49%, 41%, and 25% for laryngeal, oral, oropharyngeal and hypopharyngeal cancer patients, respectively [9]. 

A key factor contributing to poor survival rates in HNSCC patients is radiation resistance, leading to tumour recurrence [10]. The majority of HNSCC express high levels of the Epidermal Growth Factor Receptor (EGFR) which enhances DNA repair, leading to resistance [11]. In addition, as with many other tumour types, HNSCCs tend to have mutations in the tumour suppressor gene *P53*, which also results in enhanced radiation resistance [12]. Unfortunately, those patients who do not respond to irradiation will fail to be identified until they have experienced radiation doses of between 60–80 Gray (Gy), in 1.15–2 Gy fractions over a 6–7 weeks period [9]. It would be extremely useful to be able to detect patient response at the outset of therapy, as this would allow alternative treatment options to be explored, whilst minimising patient morbidity and unnecessary cost. 

Panels of genes have been identified, using cell lines, which may be associated with radio resistance. For example both You et al. [13] and Kim et al. [14] validated panels of genes associated with radiation resistance and sensitivity respectively by correlating them with outcome in patients treated with irradiation (*n* = 283 and *n* = 203 respectively). You et al. [13] identified Insulin-like Growth Factor 1 Receptor (IGF1R), Laminin Subunit Gamma 2 (LAMC2), Integrin Subunit Beta 1 (ITGB1) and Interleukin-6 (IL6) as “key” molecules associated with poor survival in HNSCC patients receiving radiotherapy, and Kim et al. [14] found that specific radio resistance signatures could predict outcome following radiation in HPV negative HNSCC, but that tumour subtype needed to be taken into consideration. Neither of these studies have yet translated into clinically useful tools. The use of cell lines to generate predictive gene signatures has limitations in that the cell lines are not direct representations of a patient’s tumour. Although cell line monolayers provide a high throughput, consistent determination of treatment response, they lack the complexity and heterogeneity of the tumour they are modelling in terms of the 3-Dimensional (3D) interactions with other cell types, including immune cells and fibroblasts, and the intricate nature of the extracellular matrix [15]. The generation of 3D spheroid models and organoids derived from different cell types is constantly being developed and goes some way to address these limitations; however, the intricacies remain challenging, as there is always the question of whether the model reflects the in vivo situation [15]. In contrast, animal models, such as patient derived xenografts and genetically engineered mouse models, provide a better, well-defined model of the cancer, but have limitations in terms of time to develop, low throughput, cost, murine influences and ethical issues [16]. The authors believe that the use of tissue-on-chip technology, where a piece of the patient’s own tumour is maintained under continuous flow, is the best representation of a model to assess a patient’s response to treatment.

The bio-microfluidics group in Hull have, over the last two decades, developed several devices which maintain human tissue ex vivo, enabling interrogation of responses to treatment [17,18,19,20,21,22,23,24,25,26]. These bespoke devices maintain the 3D architecture of the patient tissue, critical for the intercellular communication between the whole gamut of cells present. 

The biopsy of tissue is maintained for a number of days, using a dynamic flow of nutrients over the tissue with a concomitant removal of waste products, mimicking the blood and lymphatic systems in vivo (Figure 1). The Poly(Methyl Methacrylate) (PMMA/Perspex) design of the perfusion device used in the current study enables tissue to be irradiated at set points during the maintenance period, to mimic the fractionated doses received by patients. A minimum of 6 h was used between fractionation doses to allow for recovery of healthy cell function. 

The aim of the current study was to determine the efficacy of using the Hull tissue-on-chip model to determine effects of both time and clinically-relevant radiation doses on HNSCC tissue, maintained ex vivo at the global molecular and secretome level, to establish the feasibility of the technology. 

## 2. Materials and Methods

Ethical approval for the study was obtained from the National Research Ethics Service, Yorkshire and the Humber (10/H1304/6) and from Hull University Teaching Hospitals NHS Trust Research and Development (R0987). Tissue samples were obtained from patients undergoing resection surgery for removal of HNSCC; all patients were treatment-naïve. Seven patients were included in the study, 5 females and 2 males, with an age range of 60–84 years and a tumour stage ranging between T2 and T4, with and without nodal involvement (Table 1).

### 2.1. Setting Up and Running Perfusion Devices

The perfusion device was designed and manufactured in Hull out of Perspex (Kingston Plastics, Hull, UK; Figure 1). The chip comprised a central, 4 mm diameter, chamber to house the tissue, flanked by two Perspex screw-in connectors, each containing 13 holes of 0.1 mm diameter, connected via Ethylene TetraFluoroEthylene (ETFE) 1/16” tubing (1516, Kinesis Ltd., St Neots, UK). The inlet tubing was connected to a 20 mL syringe (SYR6044, BD PlastiPak™ Syringe with Luer Lock, Scientific Laboratory Supplies, Nottingham, UK) via a 0.2 μm filter (83.1826.001, Sarstedt Ltd., Leicester, UK) and a one-piece leak free connector (LS-T116-100, Mengel Engineering, Brovaenget, Denmark).

On receipt of the tissue biopsy, the sample was immediately divided using scalpels (11708353, Fisher Scientific, Loughborough, UK) into 7 pieces (~15 mg each), 6 of which were placed into separate perfusion devices within 1 h, and perfused (2 µL/min) using a calibrated pressure driven pump (PHD ULTRA™ CP Syringe Pump, 88-3015, Harvard Apparatus, Holliston, MA, USA) with Dulbecco’s Modified Eagles Medium (DMEM, D6429, Merck/Sigma-Aldrich, Dorset, UK) containing 10% (*v*/*v*) foetal bovine serum (FB-1090/500, Biosera, East Sussex, UK), penicillin/streptomycin (100 U/mL and 100 mg/mL respectively; 30-002-Cl, Corning, Flintshire, UK). The perfusion devices were incubated at 37 °C and perfused continually for 24 h. Following this period of incubation, the syringes were all disconnected from the pump and 4 of the 6 perfusion devices were secured separately to the centre of the metal shelf in a Rad Source RS2000 small animal irradiator (Rad Source, Atlanta, GA, USA). The exact position of the beam had been predetermined and marked on the metal shelf following the irradiation of a sheet of gafchromic film (Gafchromic EBT; Vertec, Reading, UK). Given the relatively large field diameters produced by the RS2000, a lead collimator was used to limit the beam size, thus reducing the scatter inside the cabinet. A self-made removable square sheet of lead 10 cm^2^, with a 2 cm diameter hole at its centre, was fitted to the roof of the RS2000 to provide this collimation (giving a beam diameter of 5 cm at the position of level 4 of the RS2000). The dose prescribed to the tissue was calibrated using ThermoLuminescent Detectors (TLD) of the Harshaw 100 H (LiF: Mg, Cu, P) variety (SNP14524, Thermo Fisher Scientific, Loughborough, UK). These were chosen due to their high sensitivity [27,28], near tissue equivalence and linearity with dose up to 20 Gy [29,30]. The energy response of these TLD is also relatively constant over the energy range used in this study [31], which is essential for in-cabinet exposures. The TLD were irradiated in a clinical Xstrahl (Xstrahl, West Midlands, UK) superficial treatment machine in the Radiotherapy Department at Castle Hill Hospital to increasing levels of dose (approx. 0.3 to 1 Gy) and compared with a secondary-standard 0.3 cc air filled, graphite walled ionisation chamber (Nuclear Enterprises type 2611B) with electrometer (Wellhofer DOSE 1). Readings were corrected for ambient temperature and pressure. This chamber/electrometer combination had been calibrated in terms of air kerma, traceable to the UK primary standard of absorbed dose for photon beams at the National Physics Laboratory (NPL, Teddington, UK). The TLD chips were processed in a Harshaw TLD 5500 reader (HARSHAWTLD5500, Thermo Fisher Scientific) and a curve of dose vs. TLD charge was derived. The TLD were then placed in the RS2000 under the appropriate experimental exposure conditions required for this study and irradiated with nominal doses of 0.5, 1, 1.5 and 2 Gy. The TLD were processed to provide corrected doses, which were subsequently used for this work.

Following the first dose of irradiation, which typically took 15 min in total, all chips were reassembled on to the syringe driver and perfusion was restarted. Two of the chips and one of the non-irradiated controls were incubated for a further 24 h (48 h in total) before removal from the syringe driver and dismantling to remove the tissue. Tissue was placed immediately into 4% (*w*/*v*) ParaFormAldehyde (PFA, 158127, Merck/Sigma-Aldrich) and fixed overnight before transfer into 70% ethanol (E/0665DF/17, Fisher Scientific). The remaining three chips were removed from the syringe driver 6 h later and 2 of the devices were irradiated again with 2 Gy as described above, before being reinstated onto the driver and incubating overnight. The same process for these 3 chips was carried out the following day and again the morning after that, so the tissue was exposed to 5 × 2 Gy of irradiation in total. The chips were again finally reassembled onto the syringe driver for a further 24 h (total 96 h) before being dismantled and the tissue placed overnight in 4% PFA. Fixed tissue was transferred to 70% ethanol and then all of the irradiated tissues and controls were dehydrated through graded alcohols (90% ethanol 30 min, 95% ethanol 30 min, 100% ethanol 30 min, 100% ethanol 50 min) and Xylene (534056, Honeywell/Fisher Scientific, 30 min, followed by 45 min in fresh Xylene) before being impregnated through two changes of Epredia™ Histoplast Paraffin (12683026, Fisher Scientific) at 55 °C for 1 h each. The tissue was finally embedded in wax blocks and allowed to fully set prior to microtome sectioning at 10 µm. 

### 2.2. RNA Extraction and Nanostring

Twenty sections (10 µm) from each sample were extracted for RNA using RNeasy DSP Formalin Fixed Paraffin Embedded kit (73604, Qiagen, Manchester, UK). The RNA isolation process included a DNase treatment step (included) and RNA was eluted in 30 µL of dH_2_O. All samples were of sufficient quantity (>10 ng/µL) and quality (A260/A280 range 1.64–2.45) to progress to Nanostring nCounter^®^ analysis. Samples with initial RNA concentration < 20 ng/µL were vacuum-dried with a Genevac™ sample concentrator (12897623, Fisher Scientific) down to 10 µL. Final RNA concentration ranged from 20.4–130.5 ng/µL. All samples were applied to the nCounter^®^ Human PanCancer Progression Panel (XT-CSO-PROG1-12, Nanostring Technologies, Inc. 530 Fairview Ave N, Seattle, WA, USA) which determines the expression of 770 genes, including 30 PanCancer reference genes (full list of genes and gene names given at https://nanostring.com/resources/ncounter-pancancer-progression-panel-gene-list/, accessed on 5 September 2023).

### 2.3. Proteome Profiler™ Array

The levels of 36 cytokines were determined in the effluent from 4 control and 4 matched irradiated samples (5 × 2 Gy) using the Human Cytokine Proteome Profiler™ array (ARY005B, R&D systems, Abingdon, UK), following the manufacturer’s guidelines. Detection antibody cocktail (15 µL) was added to 700 µL of prepared effluent (centrifuged at 300× *g* to remove cellular debris), from the 96-h time point, and the solution was made up to 1.5 ml with the buffer provided and incubated for 1 h. The reaction mixture was then added to separate membranes (previously blocked for 1 h in provided blocking buffer) in a 4-well multi-dish and incubated overnight at 4 °C with end-to-end rocking. Unbound sample was removed and the membranes washed for 3 × 10 min in 20 mL of the provided wash buffer. Streptavidin-Horse Radish Peroxidase (1:2000) was added to each membrane (2 mL/membrane), for 30 min with rocking to detect bound cytokine before further washes. Chemiluminescence using the Pierce™ Electrochemiluminescence Western Blotting Substrate (32106, Thermo Scientific) and autoradiography were used to detect the position of the dots. The density of the dots was determined using the ‘analyse gels’ function within ImageJ Fiji software (version 1.54f; open source; http://imagej.net/Fiji, accessed on 25 March 2023). The mean densitometry of each cytokine duplicate was calculated and expressed as a fraction of the average density of the positive-control dots within the membrane.

### 2.4. Statistical Analysis

The resulting RCC files were imported into the Nanostring nSolver analysis software (4.0) and raw counts were thresholded to the mean of the negative controls +2 standard deviations and were normalised using the positive controls and reference genes. Subsequent data analysis to determine the differential expression of genes between the pre-perfusion samples and those incubated on the perfusion device for either 48 h or 96 h was performed using the nCounter^®^ Advanced Analysis 2.0 plug-in (Nanostring Technologies, Seattle, WA, USA). Similar analyses were performed to determine differential expression between those samples incubated on the device for the same time period without irradiation as those incubated on the device with either 2 Gy or 10 Gy irradiation. Normalised data was exported into ClustVis (https://biit.cs.ut.ee/clustvis/, accessed on 2 February 2023) to generate heatmaps and Principal Component Analysis (PCA) plots for all genes and also for those with Benjamini–Yekutieli (BY) adjusted *p* values < 0.05 and Log_2_ fold change >1 or <−1. STRING analysis (https://string-db.org/cgi/network, accessed on 25 February 2023), inputting the significantly up and down-regulated genes separately, was used with associated Log_2_ fold change values to determine pathways of importance. Pathway scores were calculated and exported from the nCounter^®^ Advanced Analysis 2.0 plug-in and Graphpad /Prism 9 was used for statistical analysis between the pre-samples and those incubated on the chip for both 48 h and 96 h, as well as between control and irradiated samples. Due to the small sample size non-parametric comparisons were made using the paired Wilcoxon signed rank test.

Results from the Proteome Profiler™ array were displayed as a bar chart and a two-way ANOVA with Bonferroni post hoc test was used to determine the effect of radiation on cytokine release.

## 3. Results

### 3.1. The Effect of Incubation Time on Gene Expression in Ex Vivo HNSCC Tissue Maintained on the Perfusion Device 

The expression of the 740 genes in the nCounter^®^ Pan Cancer progression panel were normalized against the positive controls and 30 reference genes on the same panel using the nSolver software (4.0). A heatmap was generated to show the clustering of genes between the samples without incubation on the chip (Pre), and those incubated for both 48 h and 96 h using Euclidean distance and average linkage clustering for rows and columns (Figure 2). 

It was noted that 6 out of the 7 pre-samples clustered together (Figure 2, Green boxes; one Floor of Mouth [FOM] sample separated) with less distinct clustering between the genes expressed at the 48 h and 96 h time points. The average expression of genes within the different pathways presented in the software tended to show an overall decrease from Pre to 48 h and to a lesser degree, from 48 h to 96 h, apart from the 8 genes involved in the fibrosis pathway, which demonstrated an increase over time (Appendix A). 

Using the nCounter^®^ advanced analysis software (version number 2.0.134), the changes in gene expression between individual time points compared to the Pre-sample were investigated in more detail and those genes which were significantly Differentially Expressed (DEG) between time points are shown in Figure 1 and Figure 2. Following 48 h of culture on the perfusion device, of the genes which had a Log_2_ fold change of >1 or <−1, 6 were significantly up-regulated and 18 were significantly down-regulated (Figure 1).

Increasing the incubation time on the perfusion device to 96 h resulted in more genes being both up- and down-regulated compared with the Pre sample (Figure 2; 64 up-regulated and 57 down-regulated). Cluster analysis of the DEG (BY. *p* value < 0.05 and Log_2_ fold change > 1 and <−1) showed that there was distinct clustering between the samples prior to on-chip incubation and those tissues incubated for either 48 h or 96 h on the device (Figure 3).

Pathway enrichment analysis of the significant DEG (up and down-regulated inputted separately) using the STRING database, demonstrated strong associations following 48 h incubation of the tissue on the perfusion device between Tissue Inhibitor MetalloPeptidase inhibitor 2 (*TIMP2*), Decorin (*DCN*) and Lumican (*LUM*) as well as between SMAD family member 1 (*SMAD1*) and transforming growth factor beta receptor 2 (*TGFβR2*; Figure 4a). All of these genes were down-regulated following incubation on the perfusion device for 48 h. The Kyoto Encyclopedia of Genes and Genomes pathways (KEGG) enrichment analysis through STRING demonstrated that both the TGFβ signaling pathway and the advanced glycation end product (AGE)—Receptors (R) AGE signaling pathway in diabetic complications were significantly enriched in these down-regulated genes (Figure 5a). No pathways were significantly enriched with the up-regulated genes at 48 h (Figure 4b). Numerous associations were observed between both down (Figure 4c) and up-regulated genes (Figure 4d) following 96 h incubation on the perfusion device compared to the Pre incubation tissue. This resulted in 16 pathways being significantly enriched in the up-regulated DEG, the most significant of which were the cytokine-cytokine interaction pathway, the ExtraCellular Matrix (ECM) interaction pathway, the TGFβ signaling pathway and the Phosphoinositide 3-kinase/ Protein kinase B (PI3 Akt) signaling pathway (Figure 5b). In contrast 95 KEGG pathways were significantly enriched in the down-regulated DEG, with the most significant being the HPV infection, the pathways in cancer and the PI3 Akt signaling pathway (Top 20 shown in Figure 5b).

Comparison of the DEG in both the 48 h and the 96 h incubated samples, discovered 14 genes which had a significantly altered expression at both time points (Figure 6; 6 up-regulated and 8 down-regulated). These genes overlapped at both 48 h and 96 h as expected in the PCA analysis and were completely distinct from the Pre -samples. Enrichment analysis showed a strong association between Tumour Necrosis Factor Superfamily member 10 (*TNFSF10*), *TGFβR2* (Down-regulated) and Growth Differentiation Factor 15 (*GDF15*) (Up-regulated) and, as previously highlighted, the cytokine–cytokine receptor interaction was the most significant pathway in which these genes were involved.

The advanced nCounter^®^ analysis software (version number 2.0.134) allows analysis of the combined scores of all genes involved in 36 different pathways. Box and whisker plots generated from this data demonstrate the changes in the grouped pathway level of expression between the Pre-samples and the tissue incubated on the device. Of the 36 pathways 20 demonstrated a significant decrease in gene expression following incubation on chip for 48 h with the fibrosis pathway being the only pathway with significantly increased expression (Appendix A). 

The effect of time was consolidated by 96 h, where 32/36 pathways demonstrated an overall decrease in expression, with the fibrosis pathway again being the only one to increase further (Appendix A).

### 3.2. The Effect of Irradiation on Tissue-on-Chip Gene Expression in Ex Vivo HNSCC Tissue

No obvious clustering was evident in the tissue treated with 2 Gy of irradiation versus the control tissue, maintained on chip for the equivalent length of time without irradiation, as displayed in the heatmap and PCA plot (Figure 7a,b). Although there were no significant changes in gene expression following treatment using the adjusted *p*-value, the pathway scores analysis indicates a general trend towards increased gene expression following treatment. Only the negative regulation of angiogenesis, the positive regulation of angiogenesis and the integral to membrane pathways decreased in overall expression with 2 Gy of irradiation (Figure 7c).

Similar results were seen following 10 Gy of irradiation of the tissue. No distinct clusters were observed between control and treated tissue; however, the expression of the majority of pathways increased with treatment (Figure 8a–c). Although not significant, as with the 2 Gy treatment, both the negative regulation of angiogenesis pathways and the integral to membrane pathways showed a decrease with treatment. In addition, following 10 Gy of irradiation, the sprouting angiogenesis, ECM structure, Fibrosis and lysyl oxidase (LOX) remodelling pathways also showed a decreased trend in overall expression.

Advanced analysis demonstrated no significant changes in overall pathway scores after either 2 Gy or 10 Gy of irradiation compared to the non-irradiated control using student *t* tests for comparison.

### 3.3. The Effect of Irradiation on Cytokine Release from HNSCC Tissue Maintained on the Perfusion Device

Four samples were chosen for investigation of secreted cytokines with and without irradiation. Out of the 36 cytokines measured using the Human Cytokine Proteome Profiler (Table 2), 15 were detectable in the secretome following 96 h of incubation of the tumour tissue on the perfusion device, and eight were detected at a higher level in the secretome of the treated samples compared to the untreated, but these differences were not significant (Figure 9). Interleukin 6 (IL6), IL8, macrophage Migration Inhibitory Factor (MIF) and Serpin E1/ Plasminogen activator inhibitor-1 (PAI-1) were the cytokines released at the highest levels.

## 4. Discussion

This is the first study to investigate changes in gene expression profiles of tissue maintained on perfusion devices and whether the application of clinically-relevant doses of radiation to the tissue in the device generates further detectable changes. Previously, the group has demonstrated the ability of the ‘in-house’ designed perfusion devices to maintain pieces of human tissue ex vivo and have demonstrated changes in the tissue following the application of radiation, in terms of increased cell death and reduced proliferation [18,23,25]. The current project aimed to build on these data and investigate the global expression of a panel of 770 genes using the nCounter^®^ Pan Cancer Progression panel (Nanostring) in order to identify important changes that, firstly, are induced by incubation on the perfusion device and, secondly, those which are induced by irradiation. The nCounter^®^ Pan Cancer Progression panel was chosen over other multiplex assays as it not only provides a focussed evaluation of the expression of genes involved in cancer, but also has been demonstrated to be more sensitive and robust when using RNA samples of low abundance or low quality [32]. The authors acknowledge that the sample size in the current study is small, which is why we chose to perform a focussed assessment of genes known to be associated with cancer to provide preliminary data for genes of interest which can then be studied further using quantitative real time polymerase chain reaction.

Many models are available for studying the response of cancer cells to different insults; however, how representative these are to the in vivo tumour is still under debate [33]. There is no doubt that, although cell line models, both 2D and 3D (single cell or multicellular spheroids) generate high-throughput simulations that provide some information on drug or radiation effects, they are limited in how closely they represent the complexity of the in vivo tumour. The importance of maintaining the 3D organisation of HNSCC cells has been demonstrated using oropharyngeal squamous cell carcinoma cell lines, maintained with and without a collagen scaffold [34]. Miserochhi et al. [34] demonstrated that, despite having a reduced proliferative capacity, not only did the cells cultured on the scaffold display an increased level of markers of epithelial to mesenchymal transition and matrix interactions, but they also had a more aggressive phenotype when applied to zebra fish embryos. Notably, the HPV negative cells had the most aggressive phenotype. It is also essential to take into account the complexity of the tumour and the plethora of cells and intracellular communications that are involved in tumour growth and the ability to respond to or evade treatment. The development of organoids has attempted to capture the multicellular nature of tumours but still their complexity is relatively low and models with a constant flow of nutrients to mimic the blood supply are limited [33]. For example, similar to the current study, Engelmann et al. [35] used an organotypic model in which slices of HNC tissue were incubated on top of dermal equivalents in order to maintain tissue architecture and to mimic the extracellular matrix. Despite the lack of the dynamic delivery of nutrients and removal of waste, the authors were able to demonstrate, using 13 patient explants, the ability of the model to keep the tissue proliferating and measure responses to irradiation (*n* = 5; 5 × 2 Gy) in terms of Ki67 and cleaved caspase-3 levels, respectively. The development of microfluidic technology has been central to the development of Organ-On-a-Chip (OOC) models, in which two or more cells are co-cultured under the dynamic flow of medium and their interactions monitored (reviewed by Mattei et al. [36]). Importantly, OOC has allowed the investigation of the interaction of immune cells with cancer cells and how this changes with drug treatment [36], but still the complexity of the original tumour and the heterogeneity between tumours from different patients is not recapitulated. Animal models (xenografts and avatars) are probably the closest representation of the in vivo human currently available. However, these models are under the murine influence and the cost and time of generation outweighs their usefulness in determining response quickly, on a personal level, with added ethical considerations [37]. Differences in terms of gene expression between in vivo, ex vivo and in vitro models have been detected with more epithelial to mesenchymal transition observed in the in vivo models compared to in vitro; however, the ex vivo models possessed gene expression profiles which were more representative of the in vivo [38]. Only a single study looking at gene expression in tissue maintained on perfusion devices has been published to date, by colleagues from the University of Hull; Barry et al. found changes in gene expression in brain tissue maintained on the perfusion devices following treatment with arginine methylation inhibitors [39]. 

In the current study, the gene expression levels in the recently excised tumour were compared to those in the tissue that had been maintained on a perfusion device for 48 and 96 h. Not surprisingly, there were significant differences in the gene expression levels between fresh tissue and tissue maintained on the device, with the majority of the baseline samples clustering together on the heat map, demonstrating that they exhibit similar gene expression profiles. The similarities in gene expression of the fresh tissue samples probably reflects the fact that these are all cancers from the oral cavity and from patients of a similar age. The stage of the tumour did not appear to have an additional effect on gene expression; however, the cohort was too small to draw any statistical conclusions. There were six genes at 48 h and 64 genes at 96 h post perfusion that were significantly up-regulated, compared to 18 and 58 down-regulated respectively. This suggests that the tissue does not simply die whilst on the perfusion device, supporting previous results obtained by our group with the incorporation of bromodeoxyuridine into tissue slices maintained on the same perfusion device [25]. Of the significant DEG, *TIMP2* was found to be down-regulated and closely associated with *DCN* and *LUM* following 48 h of incubation. The down-regulation of *TIMP2* conflicts with the literature, which describes the protein as a cell stress-induced gene product, forming a complex with pro Matrix Metallopeptidase 2 (MMP2) and MMP14 generating an activation cascade resulting in active MMP2 [40]. TIMP2 protein, however, despite being thought of mainly as a tumour suppressor, has displayed promoting properties in vitro and has been linked with poorer patient survival (reviewed in 40). *DCN* was also down-regulated following 48 h on the device compared to the expression in the Pre-tissue. DCN protein is also thought to act as a tumour suppressor using various mechanisms, including the inhibition of TGFβ signalling, resulting in the inhibition of proliferation, the phosphorylation of EGFR, which subsequently results in increased *P21* expression through Mitogen-Activated Protein Kinase (MAPK) signalling, and ultimately apoptosis through the release of caspase 3 [41]. The effect of DCN protein on angiogenesis is double-edged, in that it can promote angiogenesis through promoting endothelial cell adhesion and migration, but it can also work through EGFR to up-regulate anti-angiogenic factors, such as thrombospondin-1 and TIMP3 [41]. Low *DCN* expression levels, along with increased biglycan expression, has been shown to correlate with a poorer outcome for oral squamous cell carcinoma patients [42]. The link between *DCN* and *LUM* is not surprising, as proteins from both genes belong to the leucine-rich proteoglycan family, forming part of the ECM, binding to collagen fibrils and may be involved in the organisation of collagen in the ECM [41]. The LUM protein has been shown to have both oncogenic (breast, pancreas, colorectal) and tumour suppressor (melanoma) activities [43], but to date there is no data for LUM from head and neck malignancies. 

*SMAD1* and *TGFβR2* also demonstrated significant, decreased fold changes in expression in the tissue incubated in the perfusion devices for 48 h compared to the control, and were closely associated in the STRING analysis. It is not surprising that the down-regulated expression of these genes following perfusion of the tissue is closely associated, as SMADs are proteins which, as a result of phosphorylation, transduce signals from the TGFβ receptors to the nucleus affecting gene transcription and ultimately control cell proliferation [44]. As TGFβ can act as both a tumour suppressor in healthy cells and early cancer cells, but as an oncogene in later stage malignancies, the interactions between the many SMAD proteins and the TGFβ receptors is inevitably complex and mutations in the *SMAD* gene have been associated with metastasis in HNSCC and poorer patient outcome [44]. In the HNSCC literature, it is mainly a high level of SMAD 6 protein and a low level of SMAD 2 protein that have been cited to correlate with better patient survival, with SMAD 6 potentially blocking the oncogenic effects of TGFβ [45]. No previous literature has highlighted a role for *SMAD1* in HNSCC but the decreased expression of both this and *TGFβR2* could indicate a reduced proliferative capacity of the tumour on the device. 

Enrichment analysis of all the genes demonstrating significant fold changes after 48 h of incubation demonstrated that no pathways were significantly enriched in up-regulated genes and only the TGFβ signaling pathway and the AGE-RAGE signaling pathway in diabetic complications were enriched in down-regulated genes. Interestingly, the TGFβ signaling pathway was also one of the top three pathways enriched in the up-regulated DEG after 96 h incubation, but was only the 36th most significantly enriched pathways in down-regulated genes. This up- and down-regulation of *TGFβ* associated genes probably fits with the dual role of the TGFβ protein in tumour promotion and inhibition described earlier. Previously a 7 gene, *TGFβ* associated, signature has been identified which has prognostic significance in HNSCC [46]; however, of those genes, none were significantly differentially expressed between fresh and post-perfusion tissue in the current study. 

The interaction of the AGE protein with its receptors (RAGE) has been associated with the generation of reactive oxygen species and the activation of numerous pathways, including the PI3K-Akt pathway, and has been associated with progression of cancer, including HNSCC [47,48]. The significant enrichment of the AGE-RAGE signaling pathway in diabetic complications in down-regulated genes at both 48 h and 96 h post-perfusion, as well as the PI3-Akt pathway at 96 h in the current study, is therefore somewhat surprising and could suggest that, when the tumour is on chip, the aggressive proliferative nature is subdued. However, there was also an enrichment in up-regulated genes in the PI3K-Akt pathway, suggesting conversely that some cancer progression pathways may be being stimulated. The TGFβ signaling pathway and the PI3 Akt signaling pathway are two of three highlighted pathways (Ras homologous signaling being the third) by Kidacki et al. [49], as having important modifications, and are involved in the pathogenesis, invasion and metastasis of HNSCC.

Following 96 h of incubation on the perfusion device, a multitude of genes displayed interactions following enrichment analysis, resulting in a large number of pathways being significantly enriched in both up- and down-regulated DEG. Of those pathways, in addition to the TGFβ signaling pathway discussed above, the ones that were most significantly enriched in up-regulated DEG were the cytokine–cytokine interaction pathway and the ECM receptor interaction pathway. It is not surprising that incubation of the tissue on the perfusion device stimulates genes involved in the cytokine–cytokine interaction pathway as the secretion of cytokines can have both a pro- and anti tumoural response, and they can be involved in angiogenesis, immune suppression, tumour growth and progression [50]. The cytokine–cytokine interaction pathway was one pathway that was found to be enriched in DEG between healthy and HNSCC tissue in a genomic study by Xu et al. [51] and may be indicative of the activation of tumours into survival mode. The same may be true for the up-regulated genes being enriched in the ECM-receptor interaction pathway. This pathway involves the interaction of surface molecules, including integrins and proteoglycans, whose interactions lead to changes in adhesion, migration, differentiation, proliferation and apoptosis. A bioinformatics study using the National Centre for Biotechnology Information Gene Expression Omnibus database, to identify DEG between HNSCC and healthy tissue, also found that the ECM receptor interaction pathway was significantly enriched and they highlight the importance of this pathway in tumourigenesis, invasion and metastasis [52].

A large number of pathways (*n* = 95) were also found to be enriched in down-regulated genes following 96 h of incubation on the perfusion device; in addition to the AGE-RAGE signaling pathway and the PI3K-Akt pathway discussed earlier, the top most significantly enriched pathways include those involved in the HPV infection pathway and the pathways in cancer. The HPV proteins E6 and E7 not only inhibit the *P53* and Retinoblastoma tumour suppressor genes, but they can also increase the PI3K/Akt/Mammalian target of rapamycin (mTOR) signaling pathway, which is upregulated in over 90% of HNSCC, irrespective of HPV status; this increase stimulates carcinogenesis and can lead to treatment resistance [53]. Specific inhibitors of the PI3/AKT/mTOR pathway have shown promising effects on patient outcome [51]. Therefore, the down-regulation of genes involved in these pathways, and similarly the pathways in cancer, are somewhat surprising when the tissue is incubated on the perfusion device. 

In addition to those genes that were found to have close associations following STRING analysis, Angiogenin (*ANG*) and Dicer 1, Ribonuclease III (*DICER1)* were two of the top three significantly differentially expressed genes which were up- and down-regulated, respectively, at both 48 h and 96 h post-perfusion compared to the freshly excised sample. Angiogenin is a key protein involved in neovascularization/angiogenesis [54], thus the up-regulation could indicate a survival mechanism being initiated by the tumour. *DICER1* codes for the Dicer protein, which cleaves double stranded pre miRNA into miRNA which can then go on to regulate gene expression, many miRNAs have been associated with tumorigenesis in HNSCC [55] and herefore, in contrast to *ANG*, may be an indication of the tumour becoming less active. 

Advanced nCounter^®^ analysis demonstrated that the combined scores of all genes resulted in an overall decrease in the pathway scores in 20 and 32 of the 36 pathways presented at 48 h and 96 h, respectively, with only the fibrosis pathway being significantly increased at both time points. Inflammation-related fibrosis is a hallmark of cancer, but whether this is pro- or anti-tumourigenic is not fully known [56]. Fibrosis results in the thickening and scarring of tissue and involves both fibroblasts and mesenchymal stem cells, which in association with cancer cells can control the immune environment to promote tumorigenesis [56]. The fact that the overall expression of genes in this pathway are elevated following perfusion of the HNSCC tissue on chip could indicate the response of the tumour tissue to try and regenerate itself under these conditions.

Surprisingly, no significant effects of irradiation on gene expression were observed following either 2 Gy or 10 Gy of irradiation. Despite this, there appeared to be an overall increase in expression when looking at the signature scores of pathways following both 2 Gy and 10 Gy irradiation. An increased number of patients would add power to this study; however, in addition, it must be recognised that, unlike with tumour cell lines, using fresh human tissue biopsies limits the number of experimental repeats and treatment conditions that can be set up for each patient. For example, the current set-up tried to mimic the clinical delivery of irradiation to the patient (~2 Gy, 5 times/week for 6–7 weeks to a total of 66 Gy), by delivering fractionated doses (2 Gy) with time in between (6 h), but this was done in as short a time as possible (48 or 96 h), to maximise the viability of the tissue on the chip. However, this may not have given the tissue sufficient time to respond or, conversely, immediate changes in gene expression may have been missed. These have been referred to as early and late responding genes [57]. As Eke et al. found in prostate cancer cells, radiation-induced changes in gene expression can be both fractionation and time-dependent [58]. The results of the current study are in contrast to those identified by Naghavi et al., who did a prospective study on salvage surgery specimens collected from 157 patients (96 radiotherapy naïve and 61 with radiotherapy) using an Affymetrix array with PCA and Linear models for microarray analysis and found that 251 genes displayed a significant change in expression, with a significant down-regulation in both the Wnt and Myc pathways [59]. Interestingly, *GDF15* was also found to be upregulated following irradiation and has been associated with radio-resistance/sensitivity through anti-apoptotic mechanisms. 

A recent study by Millen et al. [60] using patient derived HNSCC organoids to predict response to radiotherapy, demonstrated, on a small organoid cohort (*n* = 5), that there was an increased relapse free survival following adjuvant radiotherapy, in patients whose corresponding organoids were deemed sensitive in response to ex vivo delivered radiation. The same group demonstrated similar single nucleotide variants and Copy Number Variants (CNV) in the patient tissue and the derived organoids, but with an enrichment of CNV in the organoids. The authors state that “this is expected, as cancer organoids consist entirely of tumour cells, whereas the primary tissue still contains tumour microenvironment cells including stromal and immune cells”. The fact that organoids do have inherent differences in structure, along with the fact that they take a few weeks to generate, is why we believe the use of biopsies of patient tissue provides a more rapid, closer, representation of the in vivo state of the parent tumour. The current study used a similar number of independent samples (*n* = 7), and clearly demonstrates how fresh tumour biopsies can be studied. Although it is acknowledged that the perfusion system described herein does not currently circulate immune cells, unpublished observations have shown the presence of immune cells resident in the tissue, which ultimately could contribute to the gene expression observed. 

It should be noted that changes in gene expression, in response to irradiation and other modes of stress, do not always correlate with changes in the levels of the corresponding protein, especially over the short time frames described in the current study [61], necessitating both gene expression and protein levels to be investigated. The short time frame of up to 96 h was chosen pragmatically to ensure the tissue integrity was maintained based on previous work by the authors, and due to the limited amount of tissue available per patient. However, future irradiation studies will be designed to incorporate longer incubation times to see if changes in gene expression are observed. In the current study, the levels of cytokines synthesised and released in response to irradiation, compared to controls, was also investigated. Surprisingly, despite 15 out of the 36 cytokines investigated being detectable in the chip effluent, no significant changes in cytokine release were observed following irradiation with 5 × 2 Gy. This contrasts with studies on HNSCC cell lines, which found that irradiation (2, 5 and 10 Gy) stimulated the release of IL6 within 24 h post-irradiation from fibroblasts, which will be present in the tumour biopsy samples, as well as reducing the survival of HNSCC cells [62]. Suzuki et al. [62] also found that the migration of HNSCC cells induced by irradiated fibroblasts was mediated by IL6, although this may simply represent the differences between the 2D cell line culture and the 3D tissue where the number of fibroblasts will be lower. The secretion of cytokines into the effluent over the incubation time on the perfusion device further supports the maintenance of the functionality of the tissue ex vivo.

## 5. Conclusions

In conclusion, this report details for the first time the global expression profile of 740 genes in HNSCC tissue, whilst being maintained ex vivo. It demonstrates that although some genes do decrease in expression, there are a substantial number that significantly increase following maintenance on a bespoke perfusion device. This fact, along with the detection of secreted cytokines throughout the experimental period, provide further evidence to support the use of perfusion technology as a robust method for the monitoring of tissue ex vivo. Understanding the appropriate timescales that can be used to test potential therapeutic regimens, correlating with the in vivo situation, is the logical next step.

## Data Availability

Research data are available upon reasonable request to the principal investigators following the University of Hull’s policy.

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
