# Peer review of "Head and Neck Squamous Cell Carcinoma Biopsies Maintained Ex Vivo on a Perfusion Device Show Gene Changes with Time and Clinically Relevant Doses of Irradiation"

_cancers, 2023, doi:10.3390/cancers15184575_

Round 1

Reviewer 1 Report (Previous Reviewer 1)

The manuscript is now acceptable for the publication in the present form. 

Author Response

Many thanks for your review, we found the comments useful and we are glad that you now feel the paper is acceptable for publication in Cancers

Reviewer 2 Report (New Reviewer)

The manuscript entitled "Head and neck squamous cell carcinoma biopsies maintained ex vivo on a perfusion device show gene changes with time and clinically relevant doses of irradiation" by Green V. et al., was presented as an original article in which the authors aimed to describe the usesefull of tissue-on-chip models for molecular signature studies of HNSCC manteined ex vivo. While the paper would be of interest for a wide community working on cancer treatment and therapy, the manuscript is still preliminary, and the quality of the data does not reach the expected quality standards required for publication. Several major points need to be addressed in order to strengthen the technical rigor of the manuscript. 

Major comments for the text:

In general this study lacks novelty because the usefulness of biopsies maintained ex vivo on perfusion devices has already been demonstrated in other pathophysiological contexts and is being actively studied. Furthermore, this study suffers a small number of HNSCC cancer patients. It is known that genomic varies considerably with time and environment of sample collection, age and sex. This study not only does not take the above into account, but it also does not compare the small group of cancer samples with a healthy control group. 

The Introduction section needs to be revised. The current format of the introduction is a little bit descriptive. More novelty about this study should be described in the introduction part. The introduction lacks the social significance of studying this technology. If the author can highlight its advantages and research significance, it will improve the integrity of the article.

In the Material and Methods section it is necessary indicate informations about all reagents and equipment used (manufacturers/producers and code/catalog number). 

In the Results section be consistent with the indication of numbers of genes/pathways: use the spelled out form or numerals.

Concerning the gene nomenclature, where the authors referred to a gene, please replace it with own gene name. Moreover, gene name abbreviations must be italicized.

Please, double check and correct the typos of words/acronyms/abbreviations/tenses in the text. The reviewer suggests:

a) to capitalize the first letters of full words that define the acronyms/abbreviations

b) to specify and introduce an acronym in parentheses after the written-out form

c) once you have entered an acronym, make sure that you no longer repeat the extended form in the text

d) It must be possible to read the abstract as a stand-alone work. Therefore, all abbreviations must already be defined in the abstract

Major comments for the data:

In general, the data presented in the figures are well organized. However, the main limitation of the study is that the number of samples used is not sufficient high. 

Provide the correlation coefficients and the exact p-values for the mentioned associations in PCA analysis.

Major comments for the figures:

The size of figures is too small. Characters in some boxes are not legible in real size. Please, resize. 

Provide all tables in editable format. 

Author Response

We would  like to thank the reviewer for their comments which we found very useful and have attached a document addressing the comments in a point by point format.

Round 2

Reviewer 2 Report (New Reviewer)

The authors have addressed most of my concerns. The reviewer finds the revised version of the manuscript much improved in term of data presentation and organization. She approves the manuscript for the publication in Cancers in the current state. 

This manuscript is a resubmission of an earlier submission. The following is a list of the peer review reports and author responses from that submission.

Round 1

Reviewer 1 Report

The authors of the present work studied for the first time the gene expression of patient-derived cells cultured in a perfusion platform after irradiation.  

The manuscript looks like well written and organized. The authors have presented an interesting topic in the field of HNCs. The paper should be considered after major revisions.

1.       Some of the figures have low resolution and are very difficult to understand. The authors should increase the resolution of the figures and table 2 and 3 (expecially the volcano plots);

2.       The authors should explain better why they  used a PanCancer panel of genes than RNAseq technology;

3.       Well-defined preclinical models are needed to better reproduce the HNCs TME. Commercial cell lines cultured on common monolayer supports are in vitro systems not able to mimic the microenvironment of cancer diseases. 3D models and organ-on-chip represent some valuable research resources to reproduce the different stemness and drug sensitivity of HNC tumors. For this reason, the authors should underline these aspects through a short overview of preclinical models in the field of HNCs TME. The following references should be included in the manuscript: “Organotypic Co-Cultures as a Novel 3D Model for Head and Neck Squamous Cell Carcinoma. doi: 10.3390/cancers12082330”, “Three-dimensional collagen-based scaffold model to study the microenvironment and drug-resistance mechanisms of oropharyngeal squamous cell carcinomas. doi: 10.20892/j.issn.2095-3941.2020.0482” and “Oncoimmunology Meets Organs-on-Chip. doi: 10.3389/fmolb.2021.627454”.

4.       Limitations of the study should be included.

Reviewer 2 Report

Due to the inadequate quality of the figures, the journal titled "Head and Neck Squamous Cell Carcinoma Biopsies Maintained Ex Vivo on a Perfusion Device Show Gene Changes with Time and Clinically Relevant Doses of Irradiation" cannot be evaluated and thus should be rejected.